# Soil archives of a Fluvisol: Subsurface analysis and soil history of 1 the medieval city centre of Vlaardingen, the Netherlands - an 2 integral approach 3 4 5 7 8 9 10

Sjoerd, Kluiving<sup>1,4\*</sup>, Tim de Ridder<sup>2</sup>, Marcel van Dasselaar<sup>3</sup>, Stan Roozen<sup>4</sup> & Maarten Prins<sup>4</sup>

<sup>1</sup>Vrije Universiteit Amsterdam, Faculty of Humanities, Dept. of Archaeology, De Boelelaan 1079, 1081 HV Amsterdam, The Netherlands

<sup>2</sup>City of Vlaardingen, VLAK (Archaeology Dept., Hoflaan 43, 3134 AC Vlaardingen, The Netherlands

<sup>3</sup>Arnicon, Archeomedia 2908 LJ Capelle aan den IJssel, The Netherlands

<sup>4</sup>, Vrije Universiteit Amsterdam, Faculty of Earth and Life Sciences, Department of Earth Sciences, De Boelelaan 1085, 1081 HV Amsterdam, The Netherlands

<sup>5</sup> CLUE+ Research Institute for Culture, History and Heritage. Vrije Universiteit Amsterdam, De Boelelaan 1079, 1081 HV Amsterdam, The Netherlands

Contact author; s.j.kluiving@vu.nl 21

22

11

12 13 14

15

16 17

18

#### 23 **Abstract:**

In Medieval times the city of Vlaardingen (the Netherlands) was strategically located on the

confluence of three rivers, the Meuse, the Merwede and the Vlaarding. A church of the early 8th

century AD was already located here. In a short period of time Vlaardingen developed in the 11th

century AD into an international trading place, and into one of the most important places in the former

county of Holland. Starting from the 11th century AD the river Meuse threatened to flood the

settlement. The flood dynamics have been registered in the archives of the Fluvisols and were

recognised in a multidisciplinary sedimentary analysis of these archives.

To secure the future of these vulnerable soil archives an extensive interdisciplinary research (76

mechanical drill holes, grain size analysis (GSA), thermo-gravimetric analysis (TGA), archaeological 34 remains, soil analysis, dating methods, micromorphology, and microfauna has started in 2011 to gain

knowledge on the sedimentological and pedological subsurface of the mound as well as on the well-

preserved nature of the archaeological evidence. Pedogenic features are recorded with soil description,

micromorphological and geochemical (XRF) analysis. The soil sequence of 5 meters thickness

exhibits a complex mix of 'natural' as well as 'anthropogenic layering' and initial soil formation that 39

enables to make a distinction for relatively stable periods between periods with active sedimentation. 40

In this paper the results of this interdisciplinary project are demonstrated in a number of cross-sections

- with interrelated geological, pedological and archaeological stratification. Distinction between natural 42 and anthropogenic layering is made on the occurrence of chemical elements phosphor and potassium.

A series of four stratigraphic / sedimentary units record the period before and after the flooding

disaster. Given the many archaeological remnants and features present in the lower units, in geological

terms it is assumed that the medieval landscape was drowned while it was inhabited in the 12<sup>th</sup> century

AD. After a final drowning phase in the late  $12^{th}$  century AD, as a reaction to it, inhabitants started to 47 48

raise the surface. Within archaeological terms the boundary between natural and anthropogenic layers 49 is stratigraphically lower, so that in their interpretation the living ground was dry during the  $12^{th}$  and

50 the13<sup>th</sup> centuries AD. In the discussion the geological interpretation will be compared with alternative

archaeological visions.

#### 53 Keywords: Fluvisols, soil archives, sedimentology, archaeology, anthropogenic layers

### 57 **1. Introduction**

Since the fifties of the last century archaeological excavations in the city centre of Vlaardingen started 59 to turn the view on Vlaardingen's history at first adopted by 17<sup>th</sup> and 18<sup>th</sup> century history writers that

the old medieval city was flooded by the river Meuse (Fig. 1; De Ridder, 2002). Archaeological finds

in the research area are dated to the Middle Ages (500-1500 AD), while archaeological excavations in

the city centre south of the old church revealed the border of a medieval cemetery that has been in use

between 1000 and 1050 AD. This discovery comprised a more complete story of the medieval

- structure of Vlaardingen and made clear that today's position of the church was also the position at
- 1000 AD. Assuming that there have been no other reasons to move the church it must have been the
- same position in 726/727 AD (Koch 1970, number 2).

Despite a long period of archaeological research combined with soil observations a number of research

questions regarding the landscape and soil development still exist. Based on previous research a

number of fluvial channels are assumed to date from the Iron Age, Roman Age and Middle Ages
 spanning a period of almost 2000 years of dynamic landscape and soil development (De Ridder & van

Loon, 2007; Kluiving & Vorenhout, 2010, 2011). It is still unknown what the exact location, age of

initiation and cessation of river gullies is. Also the extent, nature and stratification of the thick

anthropogenic cover layer that underlies the old town has not been systematically researched in the

74 antihopogenic cover layer that undernes the old town has not been systematically researched in the 75 past. In general the complex interrelation between natural processes like river flow, sea level rise and

76 flooding with the cultural history of Vlaardingen and initial soil development will be addressed in this

- paper.

Archaeological research in general is dedicated to small-scale excavations or limited coring campaigns

that do not always address such complex interactions of dynamic landscape development and cultural 81 habitation. This is further enhanced by the covered and protected status of the old city of Vlaardingen,

because the narrow streets and old infrastructure do not allow large scale excavations or intensive

coring campaigns to take place. In addition currently rarely developments take place that make

archaeological research necessary following Dutch legislation.

This site evolution is based on a multi proxy approach of the soil archives in a combination and

collaboration of multiple research methodologies and correlations of heterogeneous results that is

paramount in geoarchaeological research. Standard descriptions of mechanical drill holes, grain size
 analysis (GSA), thermo-gravimetric analysis (TGA), dating of archaeological remains, soil analysis,

C14 dating methods, micromorphology, and microfauna are combined in this paper. The approach is

carried out in this paper in order to reconstruct the fluvial history and deposition of the past 3000 years

- of this region in combination with the formation of Fluvisols combined with the archaeological and
- settlement history. Fluvisols are characterised that the parent material origins from fluvial and

estuarine sedimentation, while they can be sandy, silty or clayey. The next step in soil evolution of

Fluvisols, can mean (1) transformation of sedimentary lamination in a more homogenous horizon, due 95 to bioturbation, (2) decalcification, (3) increase of soil organic carbon and (4) the translocation of clay

to bioturbation, (2) decalcification, (3) increase of soil organic carbon and (4) the translocation of clay particles from the actual Be to a (future) Bt harizon. Such accesses and identification in the second

particles from the actual Be to a (future) Bt horizon. Such processes can identify initial soil
 development during a period of landscape stability. In a next phase in pedological terms clay

development during a period of landscape stability. In a next phase in pedological terms clay
 translocation causes the formation of Luvisols. The good natural fertility of most Fluvisols and

98 translocation causes the formation of Luvisols. The good natural fertility of most Fluvisols and 99 attractive dwelling sites on river levees and on higher parts in marine landscapes were

attractive overling sites on river levees and on higher parts in marine landscapes were

100 recognized in prehistoric times (WRB, 2014).

An important asset of this study is the collaboration between geologists and archaeologists to approach

the intertwined relation between natural processes and cultural activities in an urban context. To

combine different aspects of scale as well as methods of measure against historical data (cf. van de

Biggelaar et al, 2014) has not been a straightforward task so far. The results in this study, and

107 especially the sequence of events around the historical flooding of 21 December 1163 AD (Buisman

en Van Engelen, 2000, p. 348-349; and Hoek 1973) will be further elaborated in the discussion. From

an archaeological context the boundary between natural and anthropogenic layers is interpreted at a

110 lower elevation, than based on geological arguments. This has major implications on how the

11 medieval history of Vlaardingen has to be understood. From an archaeological point of view the terp

was a safe and dry living environment, while a geological interpretation indicates that the church hill

has been regularly flooded in the  $12^{\text{th}}$  and  $13^{\text{th}}$  century during which relatively thick sediment layers

were deposited. This elementary conflict in interpretation may have an impact on other research that 115 focusses on distinguishing the contact between natural and anthropogenic layers. This could have

major implications into the research of other dikes and terps in the Holocene plains of NW Europe. In

this paper a geological analysis and interpretation will be executed, the outcome of which will be

- compared in the discussion with alternate archaeological interpretations.

### 120 121

### 2. Background

The location of Vlaardingen in the Early Holocene, around 7500 years BP, was in a tidal basin that 124 was influenced by river drainage. Around 6300 yr BP, the location changed into a wetland 125 environment with swamps and small lakes (Hijma, 2009). Between 6300 and 5000 years BP the area 126 was transformed into a peat growing environment, locally first alternating with silty clay of estuarine 127 deposits (Echteld Formation), secondly alternating with shallow marine deposits of the Wormer Layer 128 of the Naaldwijk Formation: clay with very fine sand layers (salt marshes). Due to these dynamic 129 processes the Holland Peat layer is not continuous, unlike classical Dutch Holocene stratigraphy, so 130 that in some locations the Late Holocene Walcheren layer of the Naaldwijk Formation, deposited in 131 the last 2500 years, is directly on top of the Wormer layer, while in other locations the Holland Peat 132 layer is in between these two marine layers of the Naaldwijk Formation (Hijma, 2009). In Vlaardingen 133 many cultural traces of the Iron Age and Roman Period (2750-2000 BP) have been retrieved in this 134 landscape, such as west of Vlaardingen (Vos & Eijskoot, 2009). Generally the Wormer layer can be 135 found below -3 m NAP, while the Walcheren layer is located above -3 m NAP. In the late Middle 136 Ages the actual surface was at +1 m NAP, right before the significant surface lowering due to the peat 137 exploitation. Currently that surface is indeed lowered locally to -2m NAP (Vos & Eijskoot, 2009), 138 although that surface may be higher above old gully complexes that have become inversion ridges. 139

Around 1300 years BP (700 AD) located on the point bar ridge of the 'Vlaarding' creek a church was 141 founded, that was given by Heribald to the well-known missionary Willibrord. North of it, and in a 142 later stage also around the church a settlement is originating that has been called Vlaardingen (van 143 Loon & de Ridder, 2006). Vlaardingen is one of the oldest settlement nuclei of the Western Netherlands, and grows in the 11<sup>th</sup> and 12<sup>th</sup> centuries AD into one of the most important settlements in 144 145 the county Holland. From the count's court the systematic peat exploitation around the settlement has been coordinated. In this count's capital in the second half of the 11<sup>th</sup> century AD for the first time 146 147 coins have been produced on which counts titles appear. In the year 1163 AD Vlaardingen is struck by 148 a severe flooding disaster, which had serious consequences for the settlement. Large pieces of domesticated land were lost and had to be exploited again. Vlaardingen received city rights early 13th 149 150 century that were confirmed on paper at 1273 AD. On the other hand the importance of the city decreased relative to other cities in Holland in the 13<sup>th</sup> and 14<sup>th</sup> centuries AD. The settlement grows 151 152 around the church and terp, but the expansion is limited due to the dikes, a situation that continues

- until the Industrial Revolution in the  $19^{th}$  century AD (Torremans & de Ridder, 2007).

The actual centre of the city is located on a medieval terp around the old church. This resulted 156 eventually in a mound, surface: 200 by 250 meter, built up in a 4-5 meter thick sequence of clay and 157 manure in which organic rests of former occupation are extremely well preserved, e.g. wooden posts, 158 mesh walls, but also leather objects. Recently, graves were found in the city centre, dating 1000-1050 159 AD, in which not only the wooden coffins, but also the straw that covered the deceased. In human 160 teeth DNA appeared to be well preserved, classified as the oldest in the nation, turning the church hill 161 into a large database of human DNA (De Ridder et al, 2008). Vlaardingen was a principal settlement 162 in the past. In this paper we attempt to link the rise and fall of a city like Vlaardingen with the fluvial 163 and tidal dynamics in this region and to show how important the analysis of soil archives of Fluvisols 164 can be to reconstruct landscape development.

### 3. Material and methods

In order to address complex research questions around the history of the city of Vlaardingen in relation 170 to the changing landscapes and soil profiles in this once flooded area, as well as to overcome logistical 171 problems of access to the old city, a systematic mechanical coring campaign (Macro-Core) was carried 172 out (n=76), with core diameter 5 cm. The location of cores was planned to draw specific profiles and 173 has taken place in the streets and places that were accessible (Fig. 2). Special permission has been

granted by the city council to lift the street bricks and to employ the coring device. Core depth was 175 ranging between 6 and 9 meters below street level (Table 1).

- All cores have been transported to the laboratory, where they have been cut, for standard sediment 177 description (Bosch, 2008), sampling and further laboratory analysis.

Mechanical coring has delivered samples of the subsurface in a metre scale. Mechanical coring has
caused hiatuses in coring sequence, in profile sequences these hiatuses are depicted, and in layer and
unit correlations the deposit above the hiatus is assumed to be maximised to the meter scale.

A total of 211 sediment samples were used for grain size analysis and determination of organic and 184 calcareous content. Sediment analysis was performed at the Sediment Analysis Laboratory of the VU

University Amsterdam. Grain size analysis with a Sympatec HELOS KR laser-diffraction particle

sizer was applied in order to quantify grain size distributions and make statistical comparisons and

analyses. The latter was archived through end-member analysis (cf. Weltje and Prins, 2003), which
 aims at unmixing the varying grain size distributions and identify a limited number of end-members

aims at unmixing the varying grain size distributions and identify a limited number of end-members 189 that best represent the dataset. The results can be used to distinguish between lithological units, related 190 to sediment sources or depositional mechanisms. Furthermore, thermo-gravimetric analysis with a

Leco TGA 701 was applied to quantify the organic matter and carbonate content of the sediment samples. Results of TGA and GSA analyses as well as extensive description have already been

- published elsewhere (Kluiving et al, 2014).

Micromorphology was described on 21 thin sections of 15x3 cm of 13 mechanical cores. Undisturbed samples have been air dried before being impregnated with a colourless unsaturated polyester resin.

After vaporisation of the main part of the acetone samples have been hardened by gamma radiation.
 Thin sections have been prepared from the blocks (cf. Jongerius & Heintzberger, 1975), which have

been analysed with polarising microscope with enlargements 200x (cf. (Bullock et al 1985; Courty et

- al 1989). Results of these analyses as well as extensive description have already been published(Kluiving et al, 2014).

Small volumes of sediment have been sieved on a sieve with 2mm width on multiple intervals from 28
 cores (n=67). Based on combinations of species as well as conservation status, it is assumed that
 freshwater and land animals had their habitat in local-regional areas, while the saltwater specimens
 have their provenance in the North Sea and Wadden Sea. Shells and shell rests have been analysed by
 expert knowledge which has been reported on in Kluiving et al, 2014.

- expert knowledge which has been reported on in Kluiving e
  XRF values have been sampled on 10 cores through measurements with a handheld Thermo Scientific
  Field Mate Niton XRF analyser.

Radiocarbon AMS dating is carried out in Poznańskim Laboratorium Radiowęglowym in Pozan,
Poland in 23 dates where 21 samples are dated on bulk organic material and wood, while two samples are dated on human bone material (Fig. 3).

### 4. Results

4.1 Lithofacies, sediment composition and soil characteristics

- Eight main lithofacies units can be distinguished within the studied cores based on macroscopic core
- description (colour, sedimentary structures, texture; Table 2)
- The results of grain size analyses were subjected to end-member analysis, through which four end-
- members could be distinguished (Fig. 4, Table 3). All end-members have an unimodal grain size
- distribution. End-members can be related to governing factors such as sediment source or depositional
- mechanism. However, it is difficult to identify anthropogenic actions as a depositional mechanism,
- although the provenance of specific end members may be interpreted in these units. The combined %
- of EM1 and EM2 appears to be high (> 50%) in units 4, 7-2, 7-3, and 7-1, corresponding with the
- interpretation of gully deposits from the lithology where energy levels are apparently sustainable highto carry such bedload (Table 3).
- Unit 4 consists of a rather coarse clayey sand with an EM1+2 sum of more than 50%. The unit occurs
- in the higher part of the terp in elongated lenses that reflect a cultural induced depositional mechanism (system 6). But also a few isolated lenses of unit 4 occur, which based on this association this unit can
- have a natural as well as a cultural origin.
- Unit 5 is a layer with a large component of peat, described as dark (black) with natural stones, shell
- rests, sandy and loose in packing. Various organic remains like wood and plant rests occur. Usually
- this unit is intermixed with unit 6. The unit is also sandy in nature expressed by EM1+2 % of 33%,
- and is interpreted as a cultural layer, meant to artificially raise the surface (cf. van Dasselaar, 2011).
- Unit 6 consists of black silty and sandy clay layers with a large humic content, mixed with bones and
- other archaeological remains and occurs in association with unit 5, and is interpreted as a cultural layer(cf. van Dasselaar, 2011).
- Unit 7 is a sandy clay to silt poor sand, in which three sub facies have been recognised. The first
- subfacies (7-1) has an EM1+2 proportion of 37% and a relative large proportion of EM3 of 45%. The
- other two sub facies (7-2 and 7-3) have significantly coarser signatures with EM1+2 % 56-61. This
- facies is associated with gully deposits with a natural origin in the first while the upper two sub facies
- appear to be culturally influenced, because of the coarser nature as well as the presence of artefacts.
- Unit 8 is a clay with sand lamination interpreted as point bar deposits with a high proportion of fine
- end members 3 and 4.
- Unit 9-1 is a silt poor clay interpreted as low energy flood-basin deposits dominated by EM 3 and 4 of
   more than 90%, and EM1+2% of 9 %.
- Unit 9-2 has a similar lithology but with a slightly higher EM1+2 sum of 12%, and is interpreted as a
   cultural deposit.
- Unit 10-1 is an organic silt poor clay interpreted as low energy flood-basin deposits and interpreted as
- flood basin deposits, the two sub facies distinguish a natural facies with an EM1+2 sum of 13% and a
- potential more culturally influenced sub facies 10-2 with an EM1+2 sum of 22%. Within the top parts
- of unit 10-1 at core 29 brown colours and the presence of humus staining are indicative of Fluvisol
- formation processes. Also the top of 10-1 in core 39 shows similar characteristics indicative of soilprocesses.
- Unit 11-1 is a natural peat layer, interpreted as the Hollandveen layer of the Nieuwkoop Formation,
- while the subfacies 11-2 has a relatively higher clastic content, and occurs in higher stratigraphic contexts, and is therefore interpreted as redeposited peat deposits.
- The grain size of the sediments present in the units varies over units but also reveals patterns that confirm the unit subdivision. Large differences exist between the content of coarse components (EM1 + 2 %) within units 4, 5, 7-2 and 7-3. In all of these units the sum of EM1 and 2 is very high (45-90%) for system 6 and significantly lower (2-30%) for system 3.1 (Table 3). The peaty cultural layer of unit 5 differs with a 10% EM 1+2 proportion in system 3.1, while the system 6 shows a 45% proportion.
- 5 differs with a 10% EM 1+2 proportion in system 5.1, while the system 6 shows a 45% proportion. 268 Unit 7-1 varies over systems 1, 4, and 5 reflecting gully deposits with variable flow energy, with in
- system 1 showing the lowest energy and in system 4 exhibiting the highest energy.
- 4.1.1 Thin section analysis
- In several cores to the west of the mound in the top of system 1 vegetated point bar deposits with
- charcoal remains have been interpreted that have been regularly burned (Kluiving et al, 2014). Also in
- the top of system 3 (in core 15) evidence of well-conserved plant rests with artefacts show human

- presence in combination with a C-14 date on bone of 936-1015 AD (nr. 23; Fig. 3). In core 28 in the
- top of system 1 micro-evidence of cooking rests relating to slags and hearth slags (Kluiving et al,
- 2014). Artefacts disturbing the top of the peat layer in cores 30 and 55 show the presence of an old
- surface on top of system 1 that correlates with other cores (Fig. 6A).
- 4.1.2 XRF results
- Since XRF values have been measured every 40 cm in core sections (see methods), results can best be
- incorporated by comparing with the sedimentary log (Kluiving et al, 2012). Based on this comparison
- a number of transitions in the occurence of chemical elements have been established (Table 5).
- First results show that there appears a correlation between phosphor (P) and the archaeological
- sequence. All cores show in general a significant drop in P in the measured samples going downcore.
  In cores 1, 10, 25, 30 and 35 this relation is specifically clear. In cores 1, 5 and 30 it is also observed
- In cores 1, 10, 25, 30 and 35 this relation is specifically clear. Ithat in their basal parts these cores show a slight increase in P.
- Secondly it is observed that copper (Cu) and lead (Pb) are increased as well in the upper part of the 289 cores corresponding with the P trend in the sequence.
- Lastly it appears that the potassium (K) values have more constant levels through all layers (at 0.2%), a trend which does not correlate with P, Cu or Pb.
- Based on the observed trends boundaries have been drawn that separate naturally deposited layers
- from archaeological deposits/cultural layers. In most cores more than one transition in P, Cu and/or Pb
- values is present, only two cores have a single transition from high elevated values in P, Cu and/or Pb
- from low to zero values (cores 14 and 35; Table 5). At three cores Cu, P and/or Pb values are still
- slightly elevated below the basal transitions (cores 1, 5 and 30; Table 5). The basal XRF transition
- depth, from an elevated chemical element value to absence, is in most cases at the top of system 1 orthe basal occurrence of system 3.
- 4.1.3 Results of shell analysis
- Shell rests can after analysis been split up into three categories: freshwater, saltwater and continental.
- Results indicate that we can specify two groups within the analysed shell rests: Group A shows
- exclusively freshwatershells or shell rests, with some continental shell rests (n= 17). Group B shows
- an alternation between salt water and freshwater rests, alternating with continental rests (n= 12; Table
- 6). Within group B salt water shells and rests often occur higher in the profile above freshwater and306 continental rests (Figures 5a, 6a).
- 4.1.4 Radiocarbon dating

Results of the radiocarbon dating program show a two-part division in the spread of radiocarbon dates
(Fig. 3). There is a concentration of dates in the 900-1000 AD and one in the 1050-1200 AD periods.
All radiocarbon dates are plotted in the cross sections (Figs 5, 6).

4.2 Sedimentary and (partly) anthropogenic systems

Incorporating the results from the field description and laboratory sediment analyses, the lithological
 facies of the natural deposits and cultural layers were clustered into seven lithogenetic sedimentary
 and (partly) anthropogenic systems (Table 4).

All systems have a range of lithological units and contain gully, point-bar, floodbasin and floodbasin,
organic deposits, based on their lithological characteristics (Table 3). Below these units peat and clay
deposits are observed that belong, based on their lithology and positional depth, to the Holland Peat
layer of the Nieuwkoop Formation (top 4.80-5.50 m –NAP) and the Wormer Layer of the Naaldwijk
Formation (top 5.70-6.40 m –NAP).

All natural and anthropogenic systems contain a range of lithological units (Table 2). These units
 within the systems are also depicted in the cross-sections in figures 5 and 6.

In overview it is observed that the combined number of EM1 and EM2, the coarsest fraction, coarse
and medium sand is more prominent in the higher and younger systems, e.g. systems 3.1 and 6. In

- system 1 it is striking that the interpreted flood basin clays to be deposited in medium to deep water
- contain the highest proportion coarse sediment in relation to the gully and point bar deposits (Table 3).
- In general carbonate contents of sediments analysed are relatively elevated, with higher values (up till
- 20%) in the systems 1 and 4 (Table 4). The lowest values (below 10%) occur in system 6.

System 1 sediments consist of grey to grey brown sandy clay that is interpreted as gully (unit 8) and point bar deposits (unit 7) that have a dominance of EM3, where the point bar deposits have sand layers and a significantly higher finer proportion of EM4. Grey to light grey silty clays (unit 9) are interpreted as medium-deep water floodbasin deposits with an equal dominance of EM3 and 4 and a relatively low sum EM1 + 2. Organic shallow water floodbasin deposits are interpreted from grey to dark grey silty clay with similar endmember properties as unit 9.

- System 1 is the basal unit everywhere located on top of the Holland Peat or, if eroded, directly on top 342 of the Wormer layer (fig. 5). The top of the gully deposits is most likely eroded by younger systems.
- The gully deposits of system 1 are located in the centre of Figure 5 between cores 7 and 53 flanked on
- the west by flood basin deposits between cores 8 and 14. In the north-south profile system 1 gully
- (channel and point bar deposits) is located from core 53 southward to where it is cut off by system 4
- (Fig. 6). To the north the gully deposits are bounded by flood basin deposits in core 002. In core 58
- channel deposits are observed also bounded by flood basin deposits to the north. At cores 28-55
- system 1 flood basin and peat deposits also occur at higher levels, with the top between 0,25 m NAP
- and -1, 5 m NAP, confirmed by AMS C14 and archaeological dating (Fig 3; van Dasselaar, 2011). In
- cores 29 and 30 the combined data of archaeological dating and C-14 analysis ( $1130 \pm 35$  BP, core 29, Fig.3) as well as the observation of a Fluvisol profile in unit 10 below the peat in core 29 (Fig. 6a)
- suggest a stable surface for a considerable amount of time (500-1000 years).

In system 1 TGA data of sediments show patterns of relatively high amounts of 'old carbon' in flood basin deposits (units 9 and 10), while carbonate content is increased in the 'gully' deposits (units 7 and 8; Table 4). The latter high value of carbonate content in conjunction with the presence of freshwater and continental shell rests points to a provenance of detrital carbonates transported with the Maas river to this region.

System 2 interpreted as gully sediments (unit 8) are grey silty clay deposits with sand layering, a dominance in EM3, marine shells and a relatively high proportion of EM1 +2 (~10 %). Flood basin deposits in this system consist of grey silty clay with continental shell material with dominant EM3 and relatively low EM 1+2 proportions. Light brown silty clay with detritus and peat layering is interpreted as shallow water flood basin deposits.

System 2 is much more confined and is located in the east-west profile only in the centre part, eroded 366 by younger systems elsewhere (Fig. 5). Also in the north-south profile system 2 is confined to the city 367 heart of Vlaardingen and is eroded on the south side by the Maas river, and potentially in the north 368 side by younger incisions of systems 3 and 3.1 (Fig. 6). The top of system 2 is between 0.00 and 0.50m - NAP, dated 1000-1170 AD bracketed by radiocarbon dating and archaeological remains in that 369 370 system (Kluiving et al, 2014; Dasselaar, 2011). The basal parts of system 2 suggest a Medieval age 371 (600-1000 AD) of the fluvial sediments, given by the radiocarbon and archaeological dates in cores 29 372 and 30 (Fig. 6) and core 17 (Fig. 5).

System 3 gully deposits (unit 8) consist of a grey sandy to silty clay with sand banding with a dominance of EM3 and a relatively high proportion of EM1+2 (>10%). The gully sediments contain continental and freshwater shell rests. (Dark) grey to (light) brown silty clay (10) is interpreted as organic floodbasin deposits with double the amount of EM3 vs EM4. The dark-coloured sediment with the humus/detritus banding and staining is interpreted as shallow water floodbasin deposits that

- are in this stratigraphic position often disturbed by human influence.
- System 3 with associated gully and floodplain sediments has a rather discontinuous presence in the
- ast-west profile (Fig. 5), while also in the north-south profile the system appears to be dissected by
- younger systems as well as by non-deposition due to relatively high (non-eroded) remnants of systems
- 1 and 2 (Fig. 6). Two of the system 3 gullies are located within 10 to 25 meters of the old church. The
- top of system 3 is between 0 and 0,20 m NAP in north-south profile B-B' and even between 0,20 and

- 0.50 m + NAP in the east-west profile A-A. System 3 is dated 1170-1300 AD by AMS C14 and
- relative dating (Figs 3, 5a, 6a).

System 3.1 gully deposits of sandy clay to clayey sand appear in the top of the system and are sparsely

- sampled. The dark grey to grey brown silty clay with humus staining are interpreted as shallow water 390
- floodbasin deposits. The EM4 endmember is slightly dominating over the EM3 fraction, while the 391
- anthropogenic influence is reflected in the high proportion of EM1+2 (> 20%). Dark grey silty clay 392 has a similar EM3/EM4 relation but has a lower proportion of EM1+2 of almost 10%. These
- sediments are interpreted as medium-deep water flood basin deposits. System 3.1 also has a significant 394 high amount of archaeological remains.
- System 3.1 has in both profiles a more continuous cover of which the top occurs between 1.20 and 0 m
- + NAP in the city heart, and between 0 and 2.5 m - NAP on the west side of the city heart (Fig. 5).
- System 3.1 is dated 1170-1300 AD by relative dating (Figs 5a, 6a). Within system 3.1 at some
- locations we are able to distinguish two sub-systems, system 3.1a and system 3.1b (Fig. 6). While 3.1a
- can be interpreted as an erosive phase forming gullies, 3.1b is on the contrary interpreted as i) a dike
- body and ii) as sediments raised by man to elevate the surface. While the gullies are dated around 1170AD, the dike body and raised sediments are dated after that date (1170-1300 AD).
- System 4 gully deposits (unit 7-1) are grey silty sands and have a 60% of EM1+2. Grey to light brown 404 silty clay with silt/sand banding is interpreted as point bar deposits has a dominant EM3 and an almost
- 20% of EM1+2. Flood basin deposits of system 4 have similar characteristics as other systems.
- System 4 only occurs in the southern part and incises in systems 1 and 2 as well as potentially in
- system 3 (not observed). System 4 is covered by systems 3.1 and 6; the top of this system is at 1.0 m -
- NAP, and is relatively dated older than 1170 AD (stratigraphically below system 3.1). System 4
- correlates with Maas river deposits, Echteld Formation (Fig. 6).
- System 5 gully deposits (unit 8) are grey silty clay, sand and humus banding with a dominant EM3
- and with a < 10% EM1+2. Grey sandy clay has a 33% of sum EM1 + 2 with a slightly dominant EM3
- over EM4, interpreted as point bar deposits. Grey to grey brown silty clay has a dominant EM4 414 proportion.
- System 5 only occurs in the eastern part of the research area (Fig. 5), note also the deep occurrences of
- these sediments in the northern part of the north-south profile (Fig. 6). System 5 incises deeply even
- into system 1 sediments and is covered by systems 3 (although barely) and 3.1. The stratigraphic top 418
- of system 5 is at 0.50 m NAP while the system is dated 1100-1170 AD, confirmed by AMS C14 and 419 archaeological dating.
- System 6: The (dark) grey to grey black silty clay with peat banding and humus staining which
- resembles similar interpreted shallow water flood basin deposits in previous systems has a
- significantly higher proportion of EM1+2 (~27 %). EM 3 and EM 4 are almost similar in this
- anthropogenically influenced deposit. Grey silty clay resembles deeper flood basin deposits with a
- dominant EM3 proportion, and with a ~19 % proportion of EM 1+2%. Especially the remaining
- deposits of sand, (humic) sandy clay, clayey sand and silty sand have grey to various colours, and
- extremely high proportions of EM1 +2 of 60 to 90 % (Tables 3, 4). Observations of the carbonate
- content and comparisons with lower systems show that system 6 has significantly lower carbonate 428
- values (Table 4). This may be due to the fact that soil forming processes have been going on when this
- material was exposed after it was piled up. An alternative option would be that the material from
- system 6 is transported from elsewhere with a substrate with a lower carbonate value. The very coarse 432 nature of the grain size may support the latter explanation.
- System 6 is the topmost layer and covers all lower systems with a 2.5 meter's thickness in the city
- heart in an elongated shape while at the eastern and western margins of that centre the thickness of
- system 6 is only 1 meter (Fig. 5). The relative age of this system is determined at 1300AD at the base
- until the present at the surface, interpreted as an entirely cultural system, caused by human
- interference, as opposed to the other naturally deposited systems 1, 2, and 4 that, while systems 3, 3.1
- and 5 are interpreted as minor to major influenced by human's actions.

#### 440 5. Discussion

The natural subsurface of the 'Stadshart Vlaardingen' consists of an inversion relief of a number of 442 river systems with sandy gully deposits in a chronological sequence. These river systems are underlain

- by the silt-poor calcareous clay of the Wormer layer of the Naaldwijk Formation (8000-5500 yrs BP), 444
- and the Holland Peat layer of the Nieuwkoop Formation (5500-3000 BP). The oldest (river) system 1 445 in this study is incised in both of these two formations to a depth of at least 6 m – NAP (Fig. 5).
- Micromorphological evidence has demonstrated evidence of burning (micro-charcoal remains) as well
- as slags in flood basin deposits in the top of this system between 300 and 350 cm -NAP, while also a
- few archaeological traces have been found at a deeper level in gully deposits of this system (van
- Dasselaar, 2011). The top of system 1 reaches at a few places elevations of almost 0 m NAP,
- indicative of erosive processes later on.
- The settling traces that have been found in the flood basin deposits belong to the oldest gully in the
- subsurface of Vlaardingen. System 1 correlates to the 'Hoogstad' creek system of the Vlaardingen 453 system (De Ridder & van Loon, 2007) and has a minimum date of Roman Age. Considering the fact
- that there appears to be a hiatus in deposition after system 1 of approximately 1000 years, soil
- development, i.c. Fluvisols may be expected on such a surface. In general these soils are only present
- on stable surfaces, which indicate that the top of system 1 is in fact such a surface. The observation of
- indications of a few palaeosol features might confirm this (Figs 5a, 6a). In addition the XRF results
- indicate that almost all 9 measurements have their lowest chemical element occurrence at the top of
- system 1, and that elements P. Cu, and Pb increase above this level (Table 5).

The north-south profile suggests that the gully erosion of system 5 had at least predecessors in system

- 2 and possibly also systems 1 and 3 (Fig. 6). This implies that the position of the gully shape west of
- the Vlaardingen center has been almost continuously filled with water during several stages in the last 3000 years.

System 2 is interpreted as a former river deposit only occurring in the centre of the study area, and 467 being deposited between 600 and 1170 AD, just before the late medieval storms started to threaten the 468 city. It is inferred that shortly after deposition of this system most system 2 sediments around the 469 medieval terp have been eroded and swept away during later storms and floods, explaining the now 470 isolated occurrence of these sediments. Between cores 038 and 005 in figure 6 the age of interpretation 471 of the upper part of system 2 can be disputed based on the fact of an archaeological excavation that the 472 cemetery at this location has been anthropogenically raised since 1000 AD. However in this case it is 473 tempting to test the hypothesis that the cemetery was raised by inhabitants as a reaction to the flooding 474 and sedimentation of system 2 starting in 1000 AD. It can be discussed that the lower age of 600 AD of system 2 may feed the hypothesis that in the 6<sup>th</sup> or 7<sup>th</sup> century AD renewed activity of creeks and 475 476 rivers started to make the area more attractive for habitation. Potentially a church was then built at the 477 location of Vlaardingen that was existing already in the early 8<sup>th</sup> century (Koch, 1970). Following this 478 the traces of soil formation observed at the top of system 1 suggest a relatively long stable period in 479 the order of 500 years, when no deposition or other sedimentary processes were present and soil 480 forming processes could dominate.

In system 3 many small-scale gully erosional forms occur, similar to the upper part of system 5,

indicating a reactivation at the end of the sedimentary cycle. This could be caused by high water

stands tied to storm events. Also in system 3.1 many small erosional or partly depositional traces

- (sand, sandy clay), point to stream activity in the Late Middle Ages (e.g. during storm events), with 486 the surrounding organic clays interpreted as the accompanying floodbasin deposits. In the other cores
- clearly two sedimentation cycles have been observed within system 3.1 (Kluiving et al, 2014).

In geological terms system 3 can be considered as a naturally deposited sedimentary system. This is in 490

- contrast with the case of a thick sequence in cores 55, 56, 46, 47 and 49, where archaeologists have 491
- interpreted a dike body (system 3.1b), based on the occurrence of reed packages, that have been
- generally observed in dikes in the western Netherlands. It is not unlikely that first flooding and

deposition of units 3 and 3.1 took place in the northern part of the study area, after which dammingand dike building activities became a necessity observed in system 3.1b (Fig. 6).

The interpretation of system 3.1 is also debatable between a natural deposit based on the

sedimentological characteristics or an anthropogenic cover layer based on the relatively high number

of archaeological artefacts preserved within this unit. Based on lithological characteristics a number of

gullies have been observed around the position of the old church supporting a natural origin of these

deposits. A number of distortions at the top of system 3.1 testify of human influence at this surface.
 The subdivision within system 3.1 in sub-systems 3.1a (semi-natural) and 3.1b (cultural) clearly

- observed in the north-south profile (fig. 6) might be a guide to perform more detailed analysis in the
- near future on these multiple natural and cultural systems that date roughly between 1070 and 1300
- AD. In our current interpretation the semi-natural system 3.1a has eroded the substrate until -2.5 m
- NAP and -4.5 m NAP in resp. the middle and northern parts of the B-B' cross-section (Fig. 6).

Regarding the lithological signature of the human induced layers the working hypothesis is that the
 terp layer lithology reflects the content of the immediate natural substrate. There appears to be a hiatus

in deposition after deposition of System 1 associated lithological unit sediments. The hiatus is

- supported by relative dating methods, traces of observed initial soil formation, and trends in XRF 511 analyses.
- au 512

A specific feature in this study is the comparisons between scales, while archaeology usually is

concerned on small scale excavations on the contrary geology adheres to the 500-1000 meter long

profile reconstructions. It is important to bear in mind that in the dynamic landscape history in the Late

Medieval Vlaardingen elevation differences of systems occur leaving relatively old surfaces as non-

- eroded cliffs intact at relatively high elevations, while younger systems may be incised at a lower level
- on a meter scale.519

The upper two systems below system 6, 3 and 3.1 have a stratigraphically high position with their top 521 surfaces up till 0 and 0.50 m + NAP for system 3 and between 0 and 1.20 m + NAP for system 3.1. In 522 the Late Middle Ages (1200-1500 AD) the palaeosurface for the peat area in the 'Vergulde Hand' was 523 assumed to be at approximately 1 m + NAP, which was before the considerable surface lowering due 524 to human induced peat drainage. This Late Middle Age surface for the peat area is already lowered to 525 approximately 2 meters – NAP at 2000 AD (Vos & Eijskoot, 2009). This elevation corresponds with 526 the top surface of system 3.1 at the western side of the city heart (Fig. 5).

The fact that we interpret system 3.1 as partly naturally deposited during flooding events is supported 529 by observations on grain size, archaeological dating results. The difficulty in this interpretation is that 530 the surface and upper part of system 3.1 after the flooding event has been subjected to building 531 activities, such as houses and dikes. In our current interpretation the dike in the subsurface of the 532 north-south profile (system 3.1b) has been erected after the flooding event associated with the 533 deposition of system 3.1a (Fig 6). More detailed analysis will be necessary to compare lithology, 534 trends and archaeological dating on a specific time frame, e.g. 1000-1300 AD, to sort out differences 535 in natural and cultural layers.

The discussion to classify between natural or cultural deposits is a typical interdisciplinary research 538 question. Regionally so far no comparisons have been found of city histories in a lowland environment 539 with similar research approaches. Future research will have to consider if the hypothesis that systems 540 2, 3 and 3.1 are in part naturally deposited systems can be tested positively given new archaeological, 541 historical and sedimentological research, including soil analyses on Fluvisols in this region. 542

544

### 6. Conclusions

An integrated interdisciplinary analysis of the subsoil of Vlaardingen Stadshart has delivered thefollowing key data:

- The Medieval city heart of Vlaardingen is situated on top of an old river inversion landscape • 549 that delivered opportunities for settling conditions. 550 551 • The oldest system (1) in this study correlates with the Hoogstad creek of the Vlaardingen 552 system and is relatively dated to have ended 2000 years before present (de Ridder & van 553 Loon, 2007). This relatively old river course is confirmed by the initial soil development of 554 Fluvisols that has been observed in a few cores. This is supported by the XRF analysis that 555 indicate that the elements P, Cu, and Pb increase above this system. 556 557 • The start of system 2 around 600 AD correlates with archaeological evidence of the church 558 that was present at least in 726/727 AD (Koch 1970, number 2), hypothesizing the start of the 559 Vlaardingen village after a relatively long period of stability. 560 561 • The gully shape east of the city heart has been active with water running from North to South 562 from more than 2000 years BP until 1400-1500 AD. 563 564 • The higher systems 2, 3 and 3.1, although in part intensively anthropogenically disturbed have 565 been debated in this paper as representing in part natural and anthropogenetic deposits until 566 1300 AD corresponding with the increased frequency of floods in the Late Middle Ages. 567 Future research focussing on the genesis of the surficial systems in this urban context will 568 undoubtedly contribute to this intriguing interdisciplinary research question to further unravel 569 the history of Vlaardingen. 570 571 • The upper system 6 is interpreted to have been piled up by human action starting from 1300 572 AD until the present. Premature soil formation (decalcification) may have affected the system 573 in the previous 600 years. The nature of the lithology of this anthropogenic system suggests 574 provenances originating from other places than the Stadshart. 575 576 Author contribution 577 Sjoerd Kluiving coordinated the research and wrote the manuscript. Tim de Ridder held the 578 archaeological supervision on the project and contributed in writing. Marcel van Dasselaar carried out 579 the archaeological research in Vlaardingen and contributed in writing. Stan Roozen constructed the 580 figures. Maarten Prins supplied the GSA and TGA data and contributed in writing. 581 582 583 Acknowledgments 584 Many thanks go to Richard Exaltus (Micromorphology), Lisette Kootker and Laura van der Sluis 585 (bone analysis), Kay Koster (TGA and XRF analysis), Wim Kuiper (shell rest analysis) and Steven 586 Soetens (mapping, GIS). The paper benefitted very much from the comments made by reviewers Jan 587 van Mourik, Paul Sinclair and Timothy Beach. 588 589 References 590 591 Buisman, J., en A.F.V. van Engelen, 2000: Duizend jaar weer, wind en water in de lage landen, 592 deel 1 tot 1300. 593
- Dasselaar, M. van, 2011, Archeologisch onderzoek Stadshart te Vlaardingen, ArcheoMedia rapport
- A11-009-I, Capelle aan den IJssel.

Hijma, M.P., K.M. Cohen, G. Hoffmann, A.J.F. Van der Spek & E. Stouthamer, 2009. From river valley to estuary: the evolution of the Rhine mouth in the early to middle Holocene (western Netherlands, Rhine-Meuse delta). Netherlands Journal of Geosciences 88, 1, 13-54 Hoek, C., 1973: 'De Middeleeuwen', in: T. Vos-Dahmen von Buchholz (ed.), Van steurvisser tot stedeling, Flenio, Vlaardingen, 1973, 118-146. Kluiving, S.J. & M. Vorenhout, 2010, Interdisciplinair onderzoek naar archeologie, geologie, hydrologie en conservering van het cultureel erfgoed in de ondergrond van het Stadshart van Vlaardingen, een testonderzoek, IGBA rapport 2010-01, Vrije Universiteit Amsterdam. Kluiving, S. J. & M. Vorenhout, 2011, Programma van Eisen, Stadshart te Vlaardingen, Vlaardingen. IGBA Rapport 2011-07, Vrije Universiteit Amsterdam Kluiving, S.J., K. Koster & S. Roozen, 2012. Analyse van korrelgrootte, thermogravimetrische en röntgen fluorescentie eigenschappen van sedimenten uit mechanische boorkernen in het Vlaardingen-Stadshart project. IGBA Rapport 2012-05, Vrije Universiteit Amsterdam Kluiving, S.J. M. van Dasselaar, R. Exaltus, L. Kootker, K. Koster, W. Kuiper, S. Lange, T. de Ridder, S. Roozen, L. van der Sluis, S. Soetens, 2014. Synthese Vlaardingen Stadshart. IGBA Rapport 2014-01, Vrije Universiteit Amsterdam. Loon, C. Van, en de Ridder 2007, 18. Rapport A11-009-I/Archeologisch onderzoek Stadshart te Vlaardingen\33 Koch, A.C.F., 1970: Oorkondenboek van Holland en Zeeland tot 1299, I: eind van de 7e eeuw tot 1222, Nijhoff, 's-Gravenhage, OHZ. Ridder, T. de, 2002 Waar ligt het oude Vlaardingen? Een nieuw model voor een oude stad, in: Terra Nigra, december 2002, nr. 155: 36-52. Ridder, T. de & C. Van Loon, 2007: Projectcode BC006. Het profiel van Vlaardingen. VLAK-verslag 44. Ridder, T. de, Waar ligt het oude Vlaardingen? Een nieuw model voor een oude stad, Terra Nigra nr. 155, 2002, 36-53 Ridder, T. de, E. Altena, P. de Knijff, A.H.L. Vredenbregt en H.J. Luth, 2008: De zoektocht naar de oer-Vlaardinger. In: Westerheem, special nr. 1, 2008, p. 28-38 Torremans R., de Ridder T., (red.) 2007: Bureauonderzoeken 21, Plangebied Stadshart, Vlaardingen (uitgave VLAK). Vos, P.C. & Y. Eijskoot, 2009. Geo- en archeologisch onderzoek bij de opgravingen van de Vergulde Hand West (VHW) in Vlaardingen. Deltares-rapport, 0912-0245, 160 pp. Weltje G.J., M.A., Prins, 2003: Muddled or mixed? Inferring palaeoclimate from size distributions of deep-sea clastics, Sedimentary Geology vol. 162, pp. 39-62. WRB, 2014: World reference base for soil resources, 2014. http://www.fao.org/3/a-i3794e.pdf 

FAO (2007) World Reference Base for Soil Resources, Version 2007.

| 651               |                                                                                                                                                                                                                                       |
|-------------------|---------------------------------------------------------------------------------------------------------------------------------------------------------------------------------------------------------------------------------------|
| 652               | Figures                                                                                                                                                                                                                               |
| 653<br>654<br>655 | Fig. 1 Map of Europe and Netherlands showing locations of main cities and Vlaardingen, location of study area.                                                                                                                        |
| 656<br>657<br>658 | Fig. 2 Study area within city center of Vlaardingen with locations of 76 mechanical cores and position of two cross-sections A and B                                                                                                  |
| 659               | of two cross-sections A and D.                                                                                                                                                                                                        |
| 660<br>661        | Fig. 3 AMS radiocarbon dates                                                                                                                                                                                                          |
| 662<br>663        | Fig. 4 Modelled grain-size and division of end members in Vlaardingen Stadshart.                                                                                                                                                      |
| 664<br>665<br>666 | Fig. 5a East-West (A-A') east-west cross-section of mound of Vlaardingen Stadshart and its natural subsurface. Data sheet.                                                                                                            |
| 667<br>668<br>669 | Fig. 5b East-West (A-A') east-west cross-section of mound of Vlaardingen Stadshart and its natural subsurface.                                                                                                                        |
| 670<br>671<br>672 | Fig. 6a North-South (B-B') north-south cross-section of mound of Vlaardingen Stadshart and its natural subsurface. Data sheet.                                                                                                        |
| 673<br>674<br>675 | Fig. 6b North-South (B-B') north-south cross-section of mound of Vlaardingen Stadshart and its natural subsurface.                                                                                                                    |
| 676<br>677<br>678 | Tables                                                                                                                                                                                                                                |
| 679<br>680        | Table 1: Metrical data of cores in Vlaardingen Stadshart                                                                                                                                                                              |
| 681<br>682<br>683 | Table 2 All units described in this research above the Holland Peat layer that belong to the Walcheren layer of the Naaldwijk Formation and the anthropogenous top layers.                                                            |
| 684<br>685        | Table 3; End-member data organised in units and subunits, specified by systems.                                                                                                                                                       |
| 686<br>687        | Table 4; Endmember, TGA and lithological and archaeological data organised by systems, specified per unit. Chronology by archaeological and C14 AMS dating. For interpretation of                                                     |
| 688<br>689        | depositional units see Table 2.                                                                                                                                                                                                       |
| 690<br>691<br>692 | Table 5 Depth of transitions P, Cu and/or Pb as measured by a handheld XRF analyser indicated in m down core and relative to NAP. In three cores trends of slightly increasing elements below the basal transition are indicated (*). |
| 694               | Table 6 Types of shell rests divided over cores. * indicates shell rests present in systems 3.1,                                                                                                                                      |

5, and 6.

| Number of mechanical cores, core | Depth of core below street level, in |
|----------------------------------|--------------------------------------|
| diameter 5 cm                    | m                                    |
| 60                               | 6                                    |
| 1                                | 7                                    |
| 15                               | 9                                    |
| Total: 76                        | -                                    |

| Unit | Lithology                    | Natural/cultural    | Interpretation                                                   |
|------|------------------------------|---------------------|------------------------------------------------------------------|
| 4    | Sand, clayey                 | Natural or cultural | Gully deposits or Culturally deposited*                          |
| 5    | Cultural layer (peat)        | Cultural            | Culturally deposited*                                            |
| 6    | Cultural layer (other)       | Cultural            | Culturally deposited*                                            |
| 7-1  | Clay, sandy/ Sand, silt poor | Natural             | Gully deposits                                                   |
| 7-2  |                              | Natural or cultural | Gully deposits or Culturally deposited                           |
| 7-3  |                              | Natural or cultural | Gully deposits or Culturally deposited                           |
| 8    | Clay, with sand lamination   | Natural             | Point bar deposits                                               |
| 9-1  | Clay, silt poor              | Natural             | Floodbasin deposits (medium- deep water)                         |
| 9-2  | Clay, silt poor              | Natural or cultural | Floodbasin deposits (medium- deep water) or Culturally deposited |
| 10-1 | Clay, silt poor, organic     | Natural             | Floodbasin deposits (shallow water)                              |
| 10-2 | Clay, silt poor, organic     | Natural or cultural | Floodbasin deposits (shallow water) or Culturally deposited      |
| 11-1 | Peat,                        | Natural             | Holland Peat, Nieuwkoop<br>Formation                             |
| 11-2 | Peat,                        | Natural             | Holland Peat, Redeposited                                        |
| 12   | Clay, silt poor              | Natural             | Wormer layer, Naaldwijk<br>Formation                             |

| Unit | System | EM1<br>% | EM2<br>% | EM3<br>% | EM4<br>% | n  | Sum04<br>EM1<br>and -2 |
|------|--------|----------|----------|----------|----------|----|------------------------|
| 4    | 3.1    | 12.34    | 19.32    | 58.95    | 9.40     | 2  | 31.65                  |
| "    | 6      | 6.62     | 56.48    | 3.95     | 32.95    | 3  | 63.10                  |
| 5    | 3.1    | 0.95     | 9.34     | 25.83    | 63.88    | 2  | 10.30                  |
| "    | 6      | 8.99     | 36.00    | 33.20    | 21.81    | 4  | 44.99                  |
| 7-1  | 1      | 0.00     | 7.75     | 85.33    | 6.92     | 2  | 7.75                   |
| "    | 4      | 27.01    | 32.40    | 30.99    | 9.59     | 4  | 59.42                  |
| "    | 5      | 15.86    | 16.92    | 41.84    | 25.38    | 6  | 32.78                  |
| 7-2  | 3.1    | 1.69     | 17.95\   | 72.60    | 7.76     | 2  | 19.64                  |
| "    | 6      | 16.08    | 44.48    | 17.14    | 22.30    | 13 | 60.56                  |
| 7-3  | 3.1    | 1.60     | 0.00     | 63.71    | 34.69    | 1  | 1.60                   |
| "    | 6      | 5.70     | 85.81    | 5.65     | 2.84     | 2  | 91.51                  |
| 8    | 1      | 0.22     | 6.00     | 70.33    | 23.45    | 9  | 6.22                   |
| "    | 2      | 0.26     | 10.05    | 65.98    | 23.71    | 5  | 10.30                  |
| "    | 3      | 0.00     | 12.92    | 75.31    | 11.77    | 2  | 12.92                  |
| "    | 4      | 0.00     | 19.84    | 77.61    | 2.55     | 4  | 19.84                  |
| "    | 5      | 0.24     | 9.36     | 62.11    | 28.29    | 7  | 9.60                   |
| 9-1  | 1      | 5.37     | 6.74     | 42.49    | 45.40    | 7  | 12.11                  |
| "    | 2      | 0.38     | 4.64     | 70.51    | 24.47    | 12 | 5.02                   |
| "    | 3      | 0.69     | 1.54     | 50.50    | 47.26    | 1  | 2.24                   |
| "    | 4      | 0.33     | 4.79     | 70.23    | 24.65    | 2  | 5.12                   |
| "    | 5      | 2.11     | 11.26    | 34.82    | 51.82    | 7  | 13.37                  |
| 9-2  | 3.1    | 0.53     | 9.17     | 43.28    | 47.02    | 9  | 9.69                   |
| "    | 6      | 3.26     | 16.2%    | 44.29    | 36.23    | 8  | 19.48                  |
| 10-1 | 1      | 1.78     | 3.95     | 48.49    | 45.78    | 14 | 5.73                   |
| "    | 2      | 1.37     | 0.00     | 11.42    | 87.22    | 1  | 1.37                   |
| "    | 3      | 2.71     | 9.12     | 50.91    | 37.25    | 9  | 11.83                  |
| "    | 4      | 0.00     | 10.18    | 84.47    | 5.34     | 1  | 10.18                  |
| "    | 5      | 1.32     | 15.19    | 36.31    | 47.18    | 8  | 16.51                  |
| 10-2 | 3.1    | 5.37     | 14.18    | 36.48    | 43.98    | 20 | 5.73                   |
| 11-2 | 6      | 6.19     | 21.17    | 36.65    | 35.99    | 12 | 27.36                  |
| L    |        | 1        |          |          |          |    | 1                      |

| System  | Unit | EM1 % | EM2 % | EM3 % | EM4 % | N samples | Lithology                                                    | Color                                 | LOI 330<br>gr.C. | LOI 550<br>gr. C. | (LOI 550-<br>330) gr. C. | Carbonat<br>e content<br>% | Chronolo<br>gy |
|---------|------|-------|-------|-------|-------|-----------|--------------------------------------------------------------|---------------------------------------|------------------|-------------------|--------------------------|----------------------------|----------------|
| 1       | 10-1 | 1.78  | 3.95  | 48.49 | 45.78 | 14        | Silty clay,<br>humus/detri<br>tus banding<br>and<br>staining | Grey to<br>dark<br>grey               | 4.36             | 7.86              | 3.50                     | 13.36                      | Roman/Iron Age |
| 1       | 9-1  | 0.57  | 5.21  | 46.62 | 47.59 | 7         | Silty clay                                                   | Grey to<br>light<br>grey              | 3.61             | 7.07              | 3.46                     | 12.78                      | Roman/Iron Age |
| 1       | 7-1  | 0.00  | 7.75  | 85.33 | 6.92  | 2         | Sandy clay                                                   | Grey to<br>grey<br>brown              | 1.77             | 3.31              | 1.54                     | 19.22                      | Roman/Iron Age |
| 1       | 8    | 0.22  | 6.00  | 70.33 | 23.45 | 9         | Sandy clay,<br>sand<br>layering                              | Grey to<br>grey<br>brown              | 2.06             | 4.03              | 1.97                     | 18.39                      | Roman/Iron Age |
| 2       | 10-1 | 1.37  | 0.00  | 11.42 | 87.22 | 1         | Silty clay,<br>detritus/pea<br>t banding                     | Light<br>brown                        | 8.47             | 13.52             | 5.05                     | 6.39                       | 600-1170 AD    |
| 2       | 9-1  | 0.38  | 4.64  | 70.51 | 24.47 | 12        | Silty clay,                                                  | Grey                                  | 1.71             | 3.62              | 3.30                     | 8.40                       | 600-1170 AD    |
| 2       | 8    | 0.26  | 10.05 | 65.98 | 23.71 | 5         | Silty clay,<br>sand<br>banding                               | Grey                                  | 2.30             | 4.21              | 1.91                     | 14.16                      | 600-1170 AD    |
| 3       | 10-1 | 0.73  | 2.58  | 56.94 | 39.75 |           | Silty clay,<br>humus/detri<br>tus banding<br>and<br>staining | (Dark)<br>grey to<br>(light)<br>brown | 3.67             | 6.30              | 2.63                     | 10.14                      | 1200-1300 AD   |
| 3       | 9-1  | 0.69  | 1.54  | 50.50 | 47.26 | 1         | Silty clay,                                                  | Grey                                  | 5.53             | 8.83              | 3.30                     | 8.40                       | 1200-1300 AD   |
| 3       | 8    | 0.00  | 12.92 | 75.31 | 11.77 | 2         | Sandy to<br>silty clay,<br>with<br>silt/sand<br>banding      | Grey                                  | 0.60             | 1.65              | 1.04                     | 15.36                      | 1200-1300 AD   |
| 3.<br>1 | 10-2 | 5.37  | 14.18 | 36.48 | 43.98 | 20        | Silty clay,<br>humus<br>staining                             | Dark<br>grey to<br>grey<br>brown      | 7.27             | 10.97             | 3.70                     | 8.52                       | 1200-1300 AD   |
| 3.<br>1 | 5    | 0.95  | 9.34  | 25.83 | 63.88 | 2         | Peat, mixed<br>w/clay and<br>cultural<br>remains             | Black                                 | 8.17             | 13.87             | 5.70                     | 8.55                       | 1200-1300 AD   |
| 3.      | 9-2  | 0.53  | 9.17  | 43.28 | 47.02 | 9         | Silty clay                                                   | (Dark-                                | 2.27             | 4.88              | 2.61                     | 11.73                      | 1200-1300 AD   |
| 3.<br>1 | 7-2  | 1.69  | 17.95 | 72.60 | 7.76  | 2         | Sandy clay                                                   | Dark<br>grey to<br>dark<br>brown      | 3.09             | 6.02              | 2.92                     | 10.73                      | 1200-1300 AD   |
| 3.      | 4    | 12.34 | 19.32 | 58.95 | 9.40  | 2         | Clayey                                                       | Brown                                 | 8.86             | 13.38             | 4.52                     | 10.49                      | 1200-1300 AD   |
| 3.      | 7-3  | 1.60  | 0.00  | 63.71 | 34.69 | 1         | Sandy clay                                                   | Grey                                  | 3.54             | 6.38              | 2.84                     | 12.47                      | 1200-1300 AD   |
| 4       | 10-1 | 0.00  | 10.18 | 84.47 | 5.34  | 1         | Silty clay,<br>light humus<br>content                        | Grey                                  | 2.39             | 4.51              | 2.11                     | 14.3                       |                |
| 4       | 9-1  | 0.33  | 4.79  | 70.23 | 24.65 | 2         | Silty clay                                                   | Grey                                  | 1.56             | 3.14              | 1.58                     | 19.28                      |                |
| 4       | 7-1  | 27.01 | 32.40 | 30.99 | 9.59  | 4         | Silty sand                                                   | Grey                                  | 1.78             | 3.37              | 1.59                     | 19.09                      |                |
| 4       | 8    | 0.00  | 19.84 | 77.61 | 2.55  | 4         | Silty clay,<br>silt banding                                  | Grey to<br>light<br>brown             | 1.78             | 3.37              | 1.59                     | 19.09                      |                |
| 5       | 10-1 | 1.32  | 15.19 | 36.31 | 47.18 | 8         | Silty clay,<br>humus<br>banding/<br>staining                 | Grey to<br>dark<br>brown              | 6.06             | 10.06             | 4.01                     | 7.30                       | 1100-1200 AD   |
| 5       | 9-1  | 2.11  | 11.26 | 34.82 | 51.82 | 7         | Silty clay                                                   | Grey to grey                          | 9.72             | 16.44             | 6.72                     | 7.49                       | 1100-1200 AD   |

# 707 <u>Table 4</u>

|   |      |       |       |       |                  |     |                                                      | brown                              |       |       |       |       |                 |
|---|------|-------|-------|-------|------------------|-----|------------------------------------------------------|------------------------------------|-------|-------|-------|-------|-----------------|
| 5 | 7-1  | 15.86 | 16.92 | 41.84 | 25.38            | 6   | Sandy clay                                           | Grey                               | 3.92  | 6.63  | 2.71  | 8.69  | 1100-1200 AD    |
| 5 | 8    | 0.24  | 9.36  | 62.11 | 28.29            | 7   | Silty clay,<br>sand and<br>humus<br>banding          | Grey                               | 1.90  | 3.83  | 1.93  | 17.66 | 1100-1200 AD    |
| 6 | 10-2 | 6.19  | 21.17 | 36.65 | 35.99            | 12  | Silty clay,<br>peat<br>banding,<br>humus<br>staining | (Dark)<br>grey to<br>grey<br>black | 8.23  | 13.00 | 4.77  | 7.13  | 1400 AD-present |
| 6 | 5    | 8.99  | 36.00 | 33.20 | 21.81            | 4   | Peat, mixed<br>w/clay and<br>cultural<br>remains     | Black                              | 26.47 | 45.28 | 18.81 | 5.30  | 1400 AD-present |
| 6 | 9-2  | 3.26  | 16.23 | 44.29 | 36.23            | 8   | Silty clay                                           | Grey                               | 2.27  | 4.88  | 2.61  | 11.73 | 1400 AD-present |
| 6 | 7-2  | 16.08 | 44.48 | 17.14 | 22.30            | 13  | Sand, sandy<br>clay, humic                           | Dark<br>grey                       | 7.33  | 11.70 | 4.36  | 8.66  | 1400 AD-present |
| 6 | 4    | 6.62  | 56.48 | 3.95  | 32.95            | 3   | Clayey<br>sand                                       | Variou<br>s                        | 1.87  | 3.60  | 1.73  | 6.71  | 1400 AD-present |
| 6 | 7-3  | 5.70  | 85.81 | 5.65  | 2.84             | 2   | Silty sand                                           | Variou<br>s                        | 3.93  | 6.61  | 2.68  | 6.02  | 1400 AD-present |
|   |      |       |       |       |                  |     |                                                      |                                    |       |       |       |       |                 |
|   |      |       |       |       | Total<br>samples | 186 |                                                      |                                    |       |       |       |       |                 |

| Core | Depth of transitions (/) in m down core (m +/- NAP)                 |
|------|---------------------------------------------------------------------|
| 1    | -2,55 (0,69)/-3,0 (0,23)/-4,51 (-1,28)/ *trend Cu and P             |
| 5    | -2,70 (0,58)/ -3,20 (0,08)/ -5,26 (-1,98)/ *trend Cu, P and Pb      |
| 10   | -2,37 (-2,03)/-3,40 (-3,13)/-3,75 (-3,42)                           |
| 14   | -2,45 (-3,20)                                                       |
| 20   | -2,55 (0,59)/-3,00 (0,13)/ -3,36 (-0,23) /-4,12 (-1,00)             |
| 25   | -2,00 (1,66)/ -2,40 (1,26) -3,40 (0,26)/-4,00 (-0,34)/-5,00 (-1,43) |
| 30   | -3,00 (0,66)/-4,10 (-0,44)/ -4,85 (-1,29)/-5,21 (-1,66)/ *trend P   |
| 35   | -3,33 (-3,22)                                                       |
| 40   | -2,0 (-0,95)/-2,2 (-1,15)/ -3,04 (-1,99)/ -5,0 (-3,95)              |

| Shell rest type                  | Core numbers                                                             | Group    |
|----------------------------------|--------------------------------------------------------------------------|----------|
| Freshwater (and continental)     | 1, 5, 6, 8*, 9, 10, 12, 23*, 24*, 26, 31*, 35, 37, 42, 50*, 53, and 56*. | A (n=17) |
| Saltwater (on top of freshwater) | 14, 18, 20*, 22*, 25, 29*, 30, 36, 41, 45*, 46*,<br>en 54*.              | B (n=12) |