# Peer review of "Soil archives of a Fluvisol: Subsurface analysis and soil history of 1 the medieval city centre of Vlaardingen, the Netherlands - an 2 integral approach 3 4 5 7 8 9 10"

_SOIL, 2015_

## Referee Comment (RC1) · J.M. van Mourik (Referee) · 15 Feb 2016

Soil archives of a Fluvisols: Subsurface analysis and soil history of the medieval city centre of Vlaardingen, the Netherlands - an integral approach Sjoerd, Kluiving1,4 *, Tim de Ridder2 , Marcel van Dasselaar3 , Stan Roozen4 & Maarten Prins4

This paper presents the results of a multi proxy analysis of 76 sediment cores of the sedimentary sequences underlying the present city of Vlaardingen. The specification of the applied methods is very clear, the results are used to reconstruct the development of the city of Vlaardingen in the context of the landscape evolution, dominated

by fluvial and estuarine processes. The presented figures demonstrate clearly the distinction of the 8 identified lithofacies and the sequence of 6 sedimentary systems. The main scope of the research concerns archaeology. For the reconstruction of the development of the Vlaardingen location (including the construction of the historical terp) the authors use the archives of the fluvisols. They indicate that it is difficult to make a sharp distinction between fluvial and cultural processes, resulting in 'natural' beds and cultural beds. But it is important that the reconstructed interaction between natural and cultural processes, created the parent material for the fluvisols. Pedologically, it is relevant to refer to the next step in soil development. The authors mention 'initial soil formation'. Initial soil formation, or better, the next step in soil evolution of fluvisols, can mean (1) transformation of sedimentary lamination in a more homogenous horizon, due to bioturbation, (2) decalcification, (3) increase of soil organic carbon and (4) the translocation of clay particles from the actual Be to a (future) Bt horizon. Such processes can identify initial soil development during a period of landscape stability. The first scope of the study is the devilment of the Vlaardingen site, that means archaeology. The results of the multi proxy analysis of the soil archives of the (palaeo)fluvisols is an important tool to realize this study. The EGU subdivision SRP (Soils as a Record of the Past) promotes such investigations in which soil archives analysis play an important role. That is the reason that my advice is to accept this paper for publication in the special volume of SOIL, after minor revision. Especially the definition of the properties and further initial processes of the Fluvisols (with prefix qualifiers as tidalic, umbric and suffix qualifiers as calcaric, clayic, siltic, arenic) needs some attention.

---

## Referee Comment (RC2) · P. Sinclair (Referee) · 8 Mar 2016

I find this contribution to be a very good example of multidisciplinary analysis of urban stratigraphy. It presents a coherant Array of geoscience technical analyses within a very interesting archaeological context. The research questions are clearly formulated and the analyses are well focussed as might be expected of the authors concerned. The results are clear and not insignificant.

Recently we have seen the development of Micro stratigraphic analysis in urban sites. This is often carried out without sufficient focus upon the broader contextual stratigraphy. This paper provides a very clear example of the value of straightforward multi-scalar multi-proxy analysis (76 mechanical drill holes, grain size analysis (GSA), thermo-gravimetric analysis (TGA), archaeological remains, soil analysis, dating methods, micromorphology, and microfauna demonstrating or at least arguing very strongly the complexity of human environmental interactions. The methodological approach testing the deposits prior to and after the flooding events provide reassuring diachronic detail. On reviewing the location of the boreholes it does not appear given the somewhat restricted access to drill locations, that any random stratified approach was used. However the number and spread of drill holes does support the assumption of representativity of the samples analysed.

The tables are informative and well sructured. The AMS dates are well grouped adding support to the overall argument of the paper. The illustrations also are well chosen and clear but perhaps the core sratigraphy diagrammes figs 5 a etc will not stand much reduction though this might well not be an issue in a digital publication. Its good though that they are in colour.

Overall this paper presents an excellent example of integrated approaches to complex human environmental urban contexts and I would like to see this approach applied more widely.

---

## Short Comment (SC1) · 10 Mar 2016

This article is a welcome assessment of urban archaeology in a landscape of both dynamism and quiescence. The contexts show both today's town and its medieval and earlier past buried by flooding of the Meuse River and human infill.

The paper does well to use multiple proxies from its seventy-six cores to paint a picture of a mélange of anthropogenic, flood deposits, gullies, and soil stability and instability. The samples are backed by numerous XRF estimates of soil and sediment chemistry, many laser-diffraction particle sizer assessments, twenty-three AMS dates, shell identifications as fresh- or salt-water varieties, and numerous archaeological determinations. The soil stability occurred on top of System 1 as a flooding hiatus of a millennium but the flooding returned along with gully erosion and human infill.

The article provides evidence for a large change above System 1 (Roman/ Iron Age) because the elements P, Cu, and Pb increase above this level. The article presents these data with a clear set of illustrations and tables to show their excavations and interpretations of more than ten meters of strata. Overall, this article presents a complicated puzzle of the natural processes of sea level rise, 12th and 13th Century flooding, and human adaptation by raising the city's surface.

---

## Author Comment (AC1) · 19 Apr 2016

I thank reviewers and the topical editor for their positive comments and detailed feedback, that i will rework in the paper.

---

## Author Comment (AC2) · 19 Apr 2016

I thank reviewers and the topical editor for their positive comments and detailed feedback, that i will rework in the paper.

---

## Author Comment (AC3) · 19 Apr 2016

I thank reviewers and the topical editor for their positive comments and detailed feed-back, that i will rework in the paper.

---

## Author Comment (AC4) · 6 May 2016

We thank both reviewers and the topical editor for their valuable and positive comments to our paper. Still we have used the minor comments to produce an improved version of this paper also after internal discussions between co-authors. We hope to bring with a multidisciplinary approach in an interdisciplinary setting a valuable contribution to integrated approaches to complex human environmental urban contexts. The use of Fluvisols in this research demonstrates that even soils with minor development can record valuable data of the past resulting in convincing arguments in the Vlaardingen

history.

---

## Author Response (AR1)

Comments of topical editor and replies:

Pedologically, it is
relevant to refer to the next step in soil development. The authors mention 'initial soil
formation'. Initial soil formation, or better, the next step in soil evolution of fluvisols, can
mean (1) transformation of sedimentary lamination in a more homogenous horizon,
due to bioturbation, (2) decalcification, (3) increase of soil organic carbon and (4) the
translocation of clay particles from the actual Be to a (future) Bt horizon. Such processes
can identify initial soil development during a period of landscape stability. The
first scope of the study is the devilment of the Vlaardingen site, that means archaeology.
The results of the multi proxy analysis of the soil archives of the (palaeo)fluvisols
is an important tool to realize this study. The EGU subdivision SRP (Soils as a Record
of the Past) promotes such investigations in which soil archives analysis play an important
role. That is the reason that my advice is to accept this paper for publication in the
special volume of SOIL, after minor revision.

➢ Text has been adapted according to comments

Especially the definition of the properties
and further initial processes of the Fluvisols (with prefix qualifiers as tidalic, umbric and
suffix qualifiers as calcaric, clayic, siltic, arenic) needs some attention

➢ Text has been adapted according to comments

➢ Text has been adapted according to suggestions annotated document of topical editor, see also marked document (at the end of this document) with changes in yellow
➢ Internal discussion between co-authors have led to modify the system 3.1 interpretation into an a. and b. subsystem, diversifying into a natural flood (a.) and a cultural dike construction (b.).
➢ We changed the interpreted age of system 2 into 600-1170 AD in Table 4 and in the text.
➢ We adapted the figures 5 and 6 based on the comments and internal discussion.
➢ For further textual changes we refer to the marked document.

[revised manuscript text omitted]

Table 1

| Number of mechanical cores, core diameter 5 cm | Depth of core below street level, in m |
|---|---|
| 60 | 6 |
| 1 | 7 |
| 15 | 9 |
| Total: 76 | - |

Table 2

| Unit | Lithology | Natural/cultural | Interpretation |
|---|---|---|---|
| 4 | Sand, clayey | Natural or cultural | Gully deposits or Culturally deposited* |
| 5 | Cultural layer (peat) | Cultural | Culturally deposited* |
| 6 | Cultural layer (other) | Cultural | Culturally deposited* |
| 7-1 | Clay, sandy/ Sand, silt poor | Natural | Gully deposits |
| 7-2 | | Natural or cultural | Gully deposits or Culturally deposited |
| 7-3 | | Natural or cultural | Gully deposits or Culturally deposited |
| 8 | Clay, with sand lamination | Natural | Point bar deposits |
| 9-1 | Clay, silt poor | Natural | Floodbasin deposits (medium- deep water) |
| 9-2 | Clay, silt poor | Natural or cultural | Floodbasin deposits (medium- deep water) or Culturally deposited |
| 10-1 | Clay, silt poor, organic | Natural | Floodbasin deposits (shallow water) |
| 10-2 | Clay, silt poor, organic | Natural or cultural | Floodbasin deposits (shallow water) or Culturally deposited |
| 11-1 | Peat, | Natural | Holland Peat, Nieuwkoop Formation |
| 11-2 | Peat, | Natural | Holland Peat, Redeposited |
| 12 | Clay, silt poor | Natural | Wormer layer, Naaldwijk Formation |

Table 3

| Unit | System | EM1 % | EM2 % | EM3 % | EM4 % | n | Sum EM1 and -2 % |
|------|--------|-------|-------|-------|-------|---|------------------|
| 4 | 3.1 | 12.34 | 19.32 | 58.95 | 9.40 | 2 | 31.65 |
| " | 6 | 6.62 | 56.48 | 3.95 | 32.95 | 3 | 63.10 |
| 5 | 3.1 | 0.95 | 9.34 | 25.83 | 63.88 | 2 | 10.30 |
| " | 6 | 8.99 | 36.00 | 33.20 | 21.81 | 4 | 44.99 |
| 7-1 | 1 | 0.00 | 7.75 | 85.33 | 6.92 | 2 | 7.75 |
| " | 4 | 27.01 | 32.40 | 30.99 | 9.59 | 4 | 59.42 |
| " | 5 | 15.86 | 16.92 | 41.84 | 25.38 | 6 | 32.78 |
| 7-2 | 3.1 | 1.69 | 17.95\ | 72.60 | 7.76 | 2 | 19.64 |
| " | 6 | 16.08 | 44.48 | 17.14 | 22.30 | 13 | 60.56 |
| 7-3 | 3.1 | 1.60 | 0.00 | 63.71 | 34.69 | 1 | 1.60 |
| " | 6 | 5.70 | 85.81 | 5.65 | 2.84 | 2 | 91.51 |
| 8 | 1 | 0.22 | 6.00 | 70.33 | 23.45 | 9 | 6.22 |
| " | 2 | 0.26 | 10.05 | 65.98 | 23.71 | 5 | 10.30 |
| " | 3 | 0.00 | 12.92 | 75.31 | 11.77 | 2 | 12.92 |
| " | 4 | 0.00 | 19.84 | 77.61 | 2.55 | 4 | 19.84 |
| " | 5 | 0.24 | 9.36 | 62.11 | 28.29 | 7 | 9.60 |
| 9-1 | 1 | 5.37 | 6.74 | 42.49 | 45.40 | 7 | 12.11 |
| " | 2 | 0.38 | 4.64 | 70.51 | 24.47 | 12 | 5.02 |
| " | 3 | 0.69 | 1.54 | 50.50 | 47.26 | 1 | 2.24 |
| " | 4 | 0.33 | 4.79 | 70.23 | 24.65 | 2 | 5.12 |
| " | 5 | 2.11 | 11.26 | 34.82 | 51.82 | 7 | 13.37 |
| 9-2 | 3.1 | 0.53 | 9.17 | 43.28 | 47.02 | 9 | 9.69 |
| " | 6 | 3.26 | 16.2% | 44.29 | 36.23 | 8 | 19.48 |
| 10-1 | 1 | 1.78 | 3.95 | 48.49 | 45.78 | 14 | 5.73 |
| " | 2 | 1.37 | 0.00 | 11.42 | 87.22 | 1 | 1.37 |
| " | 3 | 2.71 | 9.12 | 50.91 | 37.25 | 9 | 11.83 |
| " | 4 | 0.00 | 10.18 | 84.47 | 5.34 | 1 | 10.18 |
| " | 5 | 1.32 | 15.19 | 36.31 | 47.18 | 8 | 16.51 |
| 10-2 | 3.1 | 5.37 | 14.18 | 36.48 | 43.98 | 20 | 5.73 |
| 11-2 | 6 | 6.19 | 21.17 | 36.65 | 35.99 | 12 | 27.36 |

Table 4

| System | Unit | EM1 % | EM2 % | EM3 % | EM4 % | N samples | Lithology | Color | LOI 330 gr.C. | LOI 550 gr. C. | (LOI 550-330) gr. C. | Carbonate content % | Chronology | sumEM1 and 2 % |
|---|---|---|---|---|---|---|---|---|---|---|---|---|---|---|
| 1 | 10-1 | 1.78 | 3.95 | 48.49 | 45.78 | 14 | Silty clay, humus/detritus banding and staining | Grey to dark grey | 4.36 | 7.86 | 3.50 | 13.36 | Roman/Iron Age | 5.73 |
| 1 | 9-1 | 0.57 | 5.21 | 46.62 | 47.59 | 7 | Silty clay | Grey to light grey | 3.61 | 7.07 | 3.46 | 12.78 | Roman/Iron Age | 5.78 |
| 1 | 7-1 | 0.00 | 7.75 | 85.33 | 6.92 | 2 | Sandy clay | Grey to grey brown | 1.77 | 3.31 | 1.54 | 19.22 | Roman/Iron Age | 7.75 |
| 1 | 8 | 0.22 | 6.00 | 70.33 | 23.45 | 9 | Sandy clay, sand layering | Grey to grey brown | 2.06 | 4.03 | 1.97 | 18.39 | Roman/Iron Age | 6.22 |
| 2 | 10-1 | 1.37 | 0.00 | 11.42 | 87.22 | 1 | Silty clay, detritus/peat banding | Light brown | 8.47 | 13.52 | 5.05 | 6.39 | 1100-1200 AD | 1.37 |
| 2 | 9-1 | 0.38 | 4.64 | 70.51 | 24.47 | 12 | Silty clay, | Grey | 1.71 | 3.62 | 3.30 | 8.40 | 1100-1200 AD | 5.02 |
| 2 | 8 | 0.26 | 10.05 | 65.98 | 23.71 | 5 | Silty clay, sand banding | Grey | 2.30 | 4.21 | 1.91 | 14.16 | 1100-1200 AD | 10.30 |
| 3 | 10-1 | 0.73 | 2.58 | 56.94 | 39.75 | | Silty clay, humus/detritus banding and staining | (Dark) grey to (light) brown | 3.67 | 6.30 | 2.63 | 10.14 | 1200-1300 AD | 3.31 |
| 3 | 9-1 | 0.69 | 1.54 | 50.50 | 47.26 | 1 | Silty clay, | Grey | 5.53 | 8.83 | 3.30 | 8.40 | 1200-1300 AD | 2.24 |
| 3 | 8 | 0.00 | 12.92 | 75.31 | 11.77 | 2 | Sandy to silty clay, with silt/sand banding | Grey | 0.60 | 1.65 | 1.04 | 15.36 | 1200-1300 AD | 12.92 |
| 3.1 | 10-2 | 5.37 | 14.18 | 36.48 | 43.98 | 20 | Silty clay, humus staining | Dark grey to grey brown | 7.27 | 10.97 | 3.70 | 8.52 | 1200-1300 AD | 19.55 |
| 3.1 | 5 | 0.95 | 9.34 | 25.83 | 63.88 | 2 | Peat, mixed w/clay and cultural remains | Black | 8.17 | 13.87 | 5.70 | 8.55 | 1200-1300 AD | 10.30 |
| 3.1 | 9-2 | 0.53 | 9.17 | 43.28 | 47.02 | 9 | Silty clay | (Dark-)Grey | 2.27 | 4.88 | 2.61 | 11.73 | 1200-1300 AD | 9.69 |
| 3.1 | 7-2 | 1.69 | 17.95 | 72.60 | 7.76 | 2 | Sandy clay | Dark grey to dark brown | 3.09 | 6.02 | 2.92 | 10.73 | 1200-1300 AD | 19.64 |
| 3.1 | 4 | 12.34 | 19.32 | 58.95 | 9.40 | 2 | Clayey sand | Brown grey | 8.86 | 13.38 | 4.52 | 10.49 | 1200-1300 AD | 31.65 |
| 3.1 | 7-3 | 1.60 | 0.00 | 63.71 | 34.69 | 1 | Sandy clay | Grey | 3.54 | 6.38 | 2.84 | 12.47 | 1200-1300 AD | 1.60 |
| 4 | 10-1 | 0.00 | 10.18 | 84.47 | 5.34 | 1 | Silty clay, light humus content | Grey | 2.39 | 4.51 | 2.11 | 14.3 | | 10.18 |
| 4 | 9-1 | 0.33 | 4.79 | 70.23 | 24.65 | 2 | Silty clay | Grey | 1.56 | 3.14 | 1.58 | 19.28 | | 5.12 |
| 4 | 7-1 | 27.01 | 32.40 | 30.99 | 9.59 | 4 | Silty sand | Grey | 1.78 | 3.37 | 1.59 | 19.09 | | 59.42 |
| 4 | 8 | 0.00 | 19.84 | 77.61 | 2.55 | 4 | Silty clay, silt banding | Grey to light brown | 1.78 | 3.37 | 1.59 | 19.09 | | 19.84 |
| 5 | 10-1 | 1.32 | 15.19 | 36.31 | 47.18 | 8 | Silty clay, humus banding/ staining | Grey to dark brown | 6.06 | 10.06 | 4.01 | 7.30 | 1100-1200 AD | 16.51 |
| 5 | 9-1 | 2.11 | 11.26 | 34.82 | 51.82 | 7 | Silty clay | Grey to grey brown | 9.72 | 16.44 | 6.72 | 7.49 | 1100-1200 AD | 13.37 |

| 5 | 7-1 | 15.86 | 16.92 | 41.84 | 25.38 | 6 | Sandy clay | Grey | 3.92 | 6.63 | 2.71 | 8.69 | 1100-1200 AD | 32.78 |
|---|-----|-------|-------|-------|-------|----|-----------|------|------|------|------|------|-------|-------|
| 5 | 8 | 0.24 | 9.36 | 62.11 | 28.29 | 7 | Silty clay, sand and humus banding | Grey | 1.90 | 3.83 | 1.93 | 17.66 | 1100-1200 AD | 9.60 |
| 6 | 10-2 | 6.19 | 21.17 | 36.65 | 35.99 | 12 | Silty clay, peat banding, humus staining | (Dark) grey to grey black | 8.23 | 13.00 | 4.77 | 7.13 | 1400 AD-present | 27.36 |
| 6 | 5 | 8.99 | 36.00 | 33.20 | 21.81 | 4 | Peat, mixed w/clay and cultural remains | Black | 26.47 | 45.28 | 18.81 | 5.30 | 1400 AD-present | 44.99 |
| 6 | 9-2 | 3.26 | 16.23 | 44.29 | 36.23 | 8 | Silty clay | Grey | 2.27 | 4.88 | 2.61 | 11.73 | 1400 AD-present | 19.48 |
| 6 | 7-2 | 16.08 | 44.48 | 17.14 | 22.30 | 13 | Sand, sandy clay, humic | Dark grey | 7.33 | 11.70 | 4.36 | 8.66 | 1400 AD-present | 60.56 |
| 6 | 4 | 6.62 | 56.48 | 3.95 | 32.95 | 3 | Clayey sand | Various | 1.87 | 3.60 | 1.73 | 6.71 | 1400 AD-present | 63.10 |
| 6 | 7-3 | 5.70 | 85.81 | 5.65 | 2.84 | 2 | Silty sand | Various | 3.93 | 6.61 | 2.68 | 6.02 | 1400 AD-present | 91.51 |
| | | | | | | | | | | | | | | |
| | | | | | Total samples | 186 | | | | | | | | |

Table 5

| Core | Depth of transitions (/) in m down core (m +/- NAP) |
|---|---|
| 1 | -2,55  (0,69)/ -3,0 (0,23)/ -4,51 (-1,28)/ *trend Cu and P |
| 5 | -2,70 (0,58)/ -3,20 (0,08)/ -5,26 (-1,98)/ *trend Cu, P and Pb |
| 10 | -2,37 (-2,03)/-3,40 (-3,13)/-3,75 (-3,42) |
| 14 | -2,45 (-3,20) |
| 20 | -2,55 (0,59)/-3,00 (0,13)/ -3,36 (-0,23) /-4,12 (-1,00) |
| 25 | -2,00 (1,66)/ -2,40 (1,26) -3,40 (0,26)/-4,00 (-0,34)/-5,00 (-1,43) |
| 30 | -3,00 (0,66)/-4,10 (-0,44)/ -4,85 (-1,29)/-5,21 (-1,66)/ *trend P |
| 35 | -3,33 (-3,22) |
| 40 | -2,0 (-0,95)/-2,2 (-1,15)/ -3,04 (-1,99)/ -5,0 (-3,95) |

Table 6

| Shell rest type | Core numbers | Group |
|---|---|---|
| Freshwater (and continental) | 1, 5, 6, 8*, 9, 10, 12, 23*, 24*, 26, 31*, 35, 37, 42, 50*, 53, and 56*. | A (n=17) |
| Saltwater (on top of freshwater) | 14, 18, 20*, 22*, 25, 29*, 30, 36, 41, 45*, 46*, en 54*. | B (n=12) |